# Retrospective Analysis of Geriatric Major Trauma Patients Admitted in the Shock Room of a Swiss Academic Hospital: Characteristics and Prognosis

**DOI:** 10.3390/jcm9051343

**Published:** 2020-05-04

**Authors:** Monica Pagin, Cédric Mabire, Michael Cotton, Tobias Zingg, Pierre-Nicolas Carron

**Affiliations:** 1Department of Emergency Medicine, Lausanne University Hospital, University of Lausanne, 1011 Lausanne, Switzerland; mikeytha@gmail.com (M.C.); pierre-nicolas.carron@chuv.ch (P.-N.C.); 2Institute of Higher Education and Research in Healthcare-IUFRS, University of Lausanne, Lausanne University Hospital, 1011 Lausanne, Switzerland; cedric.mabire@chuv.ch; 3Department of Visceral Surgery, Lausanne University Hospital, University of Lausanne, 1011 Lausanne, Switzerland; tobias.zingg@chuv.ch

**Keywords:** geriatric, major trauma, mortality, trauma resuscitation

## Abstract

Increased life expectancy exposes a great number of elderly people to serious accidents, thus increasing the amount of major geriatric trauma cases. The aim of our study was to determine the profile of elderly patients undergoing major trauma, and the contributing factors predicting mortality in this specific patient group, compared to the younger population. Retrospective analysis of 1051 patients with major trauma admitted over ten years in a Level-1 trauma center was performed. Data collected were: history, nature and type of trauma; age and sex; vital signs on admission; injury severity score; shock index; comorbidities; coagulation diathesis; injury patterns; emergency measures taken; main diagnosis; mortality; and length of hospital and intensive care unit (ICU) stay. Geriatric patients admitted for major trauma have a mortality rate almost four times greater (47%) than their younger counterparts (14%) with the same ISS. According to statistical regression analysis, anti-platelet therapy (OR 3.21), NACA (National Advisory Committee for Aeronautics) score (OR 2.23), GCS (OR 0.83), ISS (OR 1.07) and age (OR 1.06) are the main factors predicting mortality. Conclusion: Geriatric major trauma patients admitted to our trauma resuscitation area have a high mortality rate. Age, GCS, ISS and NACA scores as well as anti-platelet therapy are the main factors predicting mortality.

## 1. Introduction 

Western nations including Switzerland are confronting a definite aging of their populations. The proportion of people over 65 years in Switzerland will rise from 18% in 2014 to nearly 27% thirty years later [1]. Trauma is a major cause of morbidity and mortality, despite a decrease in road traffic injuries during the last two decades [2]. In Switzerland, severe injury is the fourth cause of death between 65 and 84 years, and the fifth above 85 years, with an all-over mortality rate of 25/100,000/year [3]. In this group, falls and road crashes account for most accidents, while among adults under 65 years, road crashes, sports and urban violence take precedence [2,4].

The classic distribution of trauma mortality in time is described as tri-modal, with immediate deaths due to irreversible injury to vital organ systems (central nervous and cardiovascular), followed by severe hemorrhagic injuries within hours of arrival at the hospital, and finally late deaths related to sepsis and multi-organ failure (MOF) [5]. Recent evidence indicates a shift from this trimodal distribution to a bimodal distribution, with a drop in late deaths over time, particularly in Western countries that have implemented trauma management systems for major injuries [6]. Treatment of major trauma victims usually requires well-organized prehospital systems, and a multidisciplinary team for the initial management in the emergency room [7,8]. Organized trauma networks allow for efficient reduction of mortality in this patient group [9,10,11,12].

Due to more autonomy and a better quality of life, elderly people will be more often exposed to accidents similar to the rest of the population [13]. Compared to younger patients, the elderly have a greater morbidity and mortality for the same type of injury [14], due to the presence of comorbidities and a diminished physiologic reserve [15,16,17]. This will cause a shift in the profile of future trauma patients with particular needs that may require adaptation of current management policies and guidelines.

The literature on major trauma concerning elderly patients is relatively sparse and sometimes outdated. Most of the published studies originate from geriatric populations in the USA [18]. Several studies indicate that the most important risk factor for post-traumatic morbidity and mortality is the severity of injury [19,20,21,22], traditionally expressed by the Injury Severity Score (ISS), the Shock Index (SI), or the Glasgow Coma Scale (GCS) [23,24,25,26]. However, age, comorbidities and effects of medications are also factors influencing morbidity and mortality [16,17]. The use of anticoagulant or anti-platelet medication, frequent among elderly patients, may affect morbidity and mortality rates in major trauma, although previous studies have not shown any increase in the risk of fatal bleeding while on warfarin as long as no cerebral injury is present [27,28]. However, empirical observations suggest otherwise in geriatric major trauma.

The aim of our study was to describe the epidemiology of major trauma in our elderly population, describe their characteristics and determine the factors contributing to their mortality, compared to the younger population.

The study was approved by the State Authorities and by Lausanne University Ethics Committee (CER-VD 2016-00850).

## 2. Patients and Methods

### 2.1. Study Design and Data Description

This is a retrospective, monocentric observational study carried out in the emergency department (ED) of the Lausanne University Hospital (CHUV), in Western Switzerland, between 2005 and 2015. It is based on the ED patient registry, including details of treatment given in the field by prehospital emergency physicians and paramedics prior to admission to the trauma resuscitation area. Furthermore, details of treatment of eligible patients were completed with data acquired from the electronic patient record. In order to reduce documentation bias, data were systematically reviewed by an emergency physician who corrected any ambiguities or conflicting information, especially hand-written but also electronically captured entries.

### 2.2. Setting

The present study included patients admitted to the trauma resuscitation area of the Lausanne University Hospital emergency department. This tertiary referral hospital receives 60,000 patients annually, serving a population of 1,500,000 and is one of the 12 accredited trauma centers in Switzerland. The prehospital Emergency Medical Services (EMS) use a specific keyword-based dispatch protocol. Trained paramedics constitute the initial response on site. Prehospital emergency physicians may be primarily dispatched to the scene in case of cardiac arrest, major trauma, respiratory distress or other life-threatening emergencies; or secondarily upon request by the paramedics on site. Pre-hospital data are systematically collected in a registry. Depending on the severity of the injuries, mechanism of injuries or presence of a life-threatening situation, the patient may be triaged to the trauma resuscitation area, at the decision of the paramedics or EMS physician. In this case, a dedicated trauma team, constituted by emergency physicians, emergency nurses, surgeons and anesthesiologists is activated.

### 2.3. Patient Selection

Inclusion criteria were all blunt trauma patients over 16 years of age primarily admitted to the trauma resuscitation area of our emergency department. Penetrating injuries (firearms and stabbing)*,* transfers from outside hospitals, burn victims and patients declared dead in the field were excluded.

The following data were extracted from the EMS and hospital registries: nature, context and type of trauma; prehospital NACA (National Advisory Committee for Aeronautics) score [29]; age and sex; vital signs on presentation; ISS; GCS; SI; main comorbidities and use of anticoagulants (e.g., acenocoumarol, heparin, enoxaparin or equivalent) or anti-platelet drugs (e.g., aspirin, clopidogrel or equivalent); injury patterns, early (within 48 h) and late (in-hospital) mortality; intensive care and total hospital lengths of stay; initial cardiopulmonary resuscitation; and radiological (angio-embolization) and surgical interventions. The eight-level prehospital NACA score is used to grade the severity of illnesses or injuries, ranging from 0 (no injury) to 7 (lethal injury or disease). It is routinely used in the field and does not rely on specific clinical or biological parameters.

The included patients were subsequently divided into two groups and analyzed according to age: those below (G < 65), and those above or equal to 65 years of age (G ≥ 65).

### 2.4. Definitions

Cardiological comorbidities: hypertension, ischemic, valvular and rhythmic heart disease, hypercholesterolemia, arterial thromboembolic and aneurismal disease. Neurological comorbidities: dementia, Parkinson’s disease, stroke and epilepsy. Pulmonary comorbidities: Chronic obstructive pulmonary disease (COPD), asthma, sleep apnea syndrome and history of venous thromboembolic disease. Metabolic comorbidities: diabetes, thyroid disease, obesity, chronic renal failure and osteoporosis. Psychiatric comorbidities: depression, bipolar disorder, borderline anxiety, drug addiction and schizophrenia. Oncological comorbidities: digestive, urological, Ear, nose and throat (ENT), pulmonary, urogenital cancers, lymphoma and melanoma. Orthopedic comorbidities: osteoarthritis, prostheses (hip or knee) and disc herniation.

### 2.5. Statistical Analysis

Uni- and multivariate analyses were carried out using Stata [30]. For the initial steps, variables were summarized with appropriate descriptors (frequencies, percentages, means and SDs) and explorative measures were based on measurement levels and data distributions. Continuous variables were compared using the Student’s t-test and the Wilcoxon–Mann–Whitney test. Categorical variables were compared using the Chi-squared test and the Fischer’s exact test. In all analyses, differences were considered statistically significant at *p* <0.05 (two-tailed). A logistic regression was performed (McFadden’s pseudo-R^2^ = 0.461) to examine the relationships between mortality and clinical factors.

## 3. Results

### 3.1. Study Group Characteristics

Patient characteristics are summarized in Table 1. A total of 1051 patients were identified, of whom 806 were (G < 65), and 245 (G ≥ 65). Mean age was 37 (SD = 14) and 77 (SD = 8) years respectively. The proportion of female patients was significantly higher in the elderly group rather than the younger patient group (41% versus 21%, *p* = 0.01). The proportion of severe injuries (based on ISS ≥16 and NACA ≥ 4) was not different between the two groups.

The most frequent site of injury was in the public domain (66%) among (G < 65). In (G ≥ 65), most occurred with almost equal frequency in the public domain (45%) or at home (42%).

Younger patients (G < 65) were significantly more frequently transported by a physician-staffed ambulance than older patients (G ≥ 65) (40% versus 33%, *p* = 0.01). The proportion of severe traumatic brain injury (TBI) was significantly higher in the (G ≥ 65) (45% versus 31%, *p* = 0.01) for the same given ISS and NACA score. Shock index was low in both groups ((G < 65) 0.8 versus (G ≥ 65) 0.6, *p* = 0.01). Anticoagulants were present in 18% in the (G ≥ 65) versus 1% of patients in the (G < 65) (*p* = 0.01) and antiplatelet drugs were present in 21% in the (G ≥ 65) versus 2% of patients in the (G < 65) (*p* = 0.01).

### 3.2. Comorbidities, Hospital Length of Stay, Outcome and Mortality Analysis

Comorbidities were present in 30% of the (G < 65) and 78% of the (G ≥ 65) (*p* = 0.01). Among the (G ≥ 65), 120 (49%) had two or more comorbidities. Comorbidities are summarized in Table 2. In the (G ≥ 65), cardiovascular, neurological and oncological comorbidities were the most frequent.

Mean hospital and ICU stays were longer in the (G < 65) (17.4 and 4.5 days versus 10.9 and 3.4 days), but the difference was significant only for hospital length of stay (*p* = 0.01). After the main hospital stay, 38% of the (G < 65) returned home, 26% went to rehabilitation, 21% were transferred to another hospital and 1.5% went to a psychiatric hospital. In contrast, in the (G ≥ 65), only 11% returned home, 17% went to rehabilitation, 24% were transferred to another hospital and 0.4% went to a psychiatric hospital (Figure 1).

Mortality was higher in the (G ≥ 65), both in the first 48 h (34% vs. 10%, *p* = 0.01) and during the hospital stay (47% vs. 14%, *p* = 0.01). Risk factors for 48 h and in-hospital mortality were anti-platelet therapy, greater age, lower GCS and higher NACA or ISS scores (Table 3).

### 3.3. Trauma Mechanism, Iinjuries Sustained and Need for Resuscitation

Fall was the main mechanism in the (G ≥ 65) (*n* = 150; 61%) and road traffic accident in the (G <65) (*n* = 392; 49%).

The three most frequent injuries encountered concerned the head (47%), chest (39%) and lower limbs (33%) (Table 4).

Two thirds of the patients (*n* = 697; 66%) underwent immediate treatment—mostly orthopedic (*n* = 269; 25%) and neurosurgical (*n* = 235; 22%) procedures. Interventions in general were more frequently performed in the (G < 65), especially for urgent orthopedic operations (Table 5).

## 4. Discussion

To our knowledge, this is the first study analyzing the medical presentation and hospital outcomes of geriatric patients admitted to an urban, academic trauma center ED in Switzerland for major trauma. In this 10-year retrospective study, elderly patients (G ≥ 65) admitted for major trauma had a three to four times higher mortality (47%) than younger counterparts (14%) with a comparable ISS. Anti-platelet therapy, NACA score, initial GCS, ISS and age were risk factors for mortality. Several previous studies [7,18,22,23,24,25,26,31,32,33] have described advanced age, along with higher ISS and lower GCS as being associated with trauma mortality.

The epidemiology and patterns of severe injuries in Switzerland and the area of Lausanne University Hospital in particular have been described by Heim et al. [34] in 2014. In their study, victims were mainly males in the early forties, with a rate of elderly trauma patients similar to reports from North American trauma centers. Blunt trauma caused 91% of injuries, mainly of the brain, chest and extremities. In comparison to other high-income countries, the rate of road traffic accidents was higher (40%) than in the United Kingdom (33%) or the Unites States (35%), but lower than in Germany (56%). Compared to data from the German Trauma Registry, more severe head (68% vs. 54%) and pelvic/extremity (40% vs. 31%) injuries were observed at our institution. This may be related to the high rate (28%) of motorbike-related incidents among road traffic crashes. Hasler et al. [33], based on data from another Swiss university hospital, also found older age, higher ISS and lower GCS to be significant predictors of mortality, in line with the results of our study.

Trauma patients with ongoing anti-platelet therapy were three times more likely to die than those without in the present series. To our knowledge, this has not been shown before. Since anti-platelet therapy is frequently present in elderly patients owing to cardiovascular and neurological disease, the impact of these agents on the risk of an unfavorable outcome due to hemorrhage, especially in cases of a severe TBI is potentially significant. In our study, 6.6% of patients were under anti-platelet therapy. Unlike some other studies [27,28,35], anticoagulation was not significantly associated with mortality in our study, potentially due to the low number of observations.

The NACA score also had a strong association with mortality in our study. This may be due to the presence of relevant variables such as comorbidities, age and treatment contributing to the final NACA score. Darioli et al. [29] showed that the discriminative performance of the NACA score was better for trauma than non-trauma cases, but decreased with increasing age due to comorbidities and multiple medications, especially beta-blockers. The frequent use of beta-blockers among the elderly may be the reason why our results show no association between shock index and mortality, in contrast to the study by Pandit et al. [26]

In our study, 23% of all major trauma patients admitted to the trauma resuscitation area were older than 64 years. In absolute numbers and independently of age, males were more frequently implicated in major trauma. These findings concord with a study by Grazlja et al. [13], but conflict with two others by Grossman et al. [16] and Giannoudis et al. [23], which counted more females in the elderly trauma population. We observed a lower absolute number, but a significantly higher proportion of women in the older than in the younger group. The tendency towards high-risk activities among men in general and the higher life expectancy in women may in part explain our observations.

As far as mechanisms of injury are concerned, road crashes dominate in the younger group, and falls in the elderly; a finding which is in line with other studies [13,18]. Physiological changes such as poor vision and unstable balance are risk factors for falls in the elderly. The different mechanisms may also be explained by dissimilar age-dependent daily activities (work, commute, sports).

The most commonly injured body regions in both groups were the head and the chest. Similar observations have previously been made [18,19]. Emergency orthopedic and abdominal interventions were more frequently performed in the younger than the older group, presumably because of more high-velocity injuries among the former. However, patients who underwent no emergent intervention at all were significantly more frequent in the older group. This may be because certain interventions were considered to be futile with increasing age, lesions such as severe TBI were present [36] or because it may illustrate some kind of “ageism”. Another explanation for a delay in treatment may be due to an underestimation of injury severity. Several studies have shown that elderly trauma patients are at risk for undertriage [37,38,39].

The delay in treatment may in turn be one of the reasons for the higher mortality among older patients. No specific age-dependent management protocols for the elderly exist at our institution. However, several studies show that outcomes may be favorably influenced by diagnostic and therapeutic protocols and multidisciplinary care adapted to elderly trauma victims [8,40,41,42].

Older patients were also significantly less frequently transported by a physician-staffed ambulance than younger patients. This may be due to the presence of an advance directive, or again, due to a limitation of care or underestimation of the injury severity. In the same population, Tavares et al. [43] observed that the number of out-of-hospital advanced medical interventions significantly decreased for patients over 89 years. According to a recent study [44], 24% of the Swiss population older than 55 years have completed an advance directive.

Mean hospital and intensive care unit stays were longer in the younger than in the older group. More serious injuries that are survivable in younger patients, but can cause death in frail elderly patients despite maximal ICU care, may explain this finding, as in the study by Grzalia et al. [13] Limitation of care after discussion with the family because of a dismal prognosis may be another reason. Verbeek et al. [32] observed shorter hospital stays for patients over 65 years with pelvic fractures compared to those under 65, likely due to the larger proportion of minor injuries resulting from low-energy mechanisms in the older group. Others showed the opposite results: McKevitt et al. [14] demonstrated that elderly polytrauma patients needed longer hospital care because of more frequent and more severe complications than those seen in younger patients.

An observation worth noting is that of the elderly, 17% and 24% were transferred in a second step respectively to other hospitals for rehabilitation. In Switzerland, the initial treatment of a severely injured patient is carried out in a limited number of trauma centers. After stabilization, patients are usually transferred to the patient’s regional hospital or to a specialized rehabilitation facility. Broos et al. [19] reported that 80% of elderly patients were at home six months after injury. These results are in contrast to our experience, where only 11% could return to their homes.

In addition to the irregularities of comparing data between different nations, some of the differences between our observations and those found in the literature may be due to the fact that part of the comparative data are quite remote and therefore changes in care provision, drug refinements (e.g., direct oral anticoagulants) and service restructure may all be factors to consider.

### Limitations and Strengths

Our study has several limitations. It is a retrospective study with the risk of documentation bias. The registry-based data were not specifically collected to answer the study questions. Mortality was compared according to age and matched for ISS, but not for all variables. No specific organ injury or subgroup analysis of injury severity or age subgroups was performed separately. This is a single-center study, reflecting local practice with limited external validity. There may be a selection bias because only patients admitted to the trauma resuscitation area were included. It is therefore possible that some patients with an ISS >15, treated in a standard ED bay, are not included in our cohort. Burn victims and victims of penetrating trauma were excluded, because of the uncommon number of cases and their management in a different pathway (burn patients), precluding any specific analysis in these subgroups. Finally, our logistic regression analysis explains only 46% of mortality. The strengths of the study are the fusion of pre- and in-hospital data and, for Swiss standards, the high numbers of patients.

## 5. Conclusions

Patients over 65 years old admitted to our trauma resuscitation area have a three to four times higher mortality (47%) than their younger counterparts (14%) with the same ISS. Anti-platelet therapy, NACA score, GCS, ISS and age are the principal risk factors for mortality among severely injured elderly patients.

## Figures and Tables

**Figure 1 jcm-09-01343-f001:**
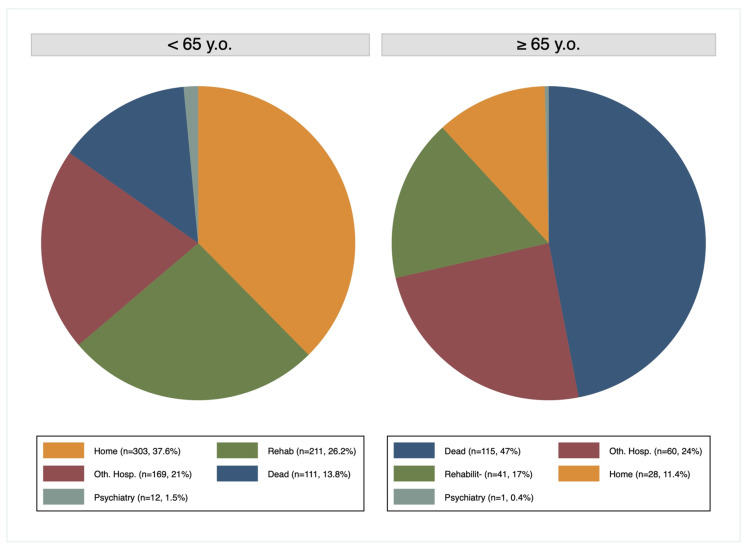
Outcome.

**Table 1 jcm-09-01343-t001:** Patient Characteristics (*n* = 1051).

	<65 years	≥65 years	Total	*p*-Value	*p*-Value
Total number of patients *n* (%)	806 (76.7)	245 (23.3)	1051		0.01 ^e^
Age mean (SD)	36.9 (14.0)	77.1 (8.0)	46.2 (21.3)	0.01 ^a^	
Male *n* (%)	640 (79.3)	144 (58.8)	784 (74.5)	0.01 ^b^	
Site of accident *n* (%)					
Public domain	532 (65.9)	110 (44.9)	642 (61.0)	0.01 ^c^	0.01 ^e^
Home	66 (8.2)	104 (42.4)	170 (16.2)		0.01 ^e^
Work place	93 (11.5)	6 (2.3)	99 (9.4)		0.45 ^e^
Hospital/Medical center	7 (0.9)	2 (0.8)	9 (0.9)		0.98 ^e^
Nursing home	2 (0.3)	6 (2.5)	8 (0.8)		0.84 ^e^
Other	107 (13.3)	17 (6.9)	124 (11.8)		0.45 ^e^
Helicopter transport *n* (%)	343 (54.3)	85 (50.9)	428 (40.7)		0.57 ^e^
Medicalized ambulance *n* (%)	254 (40.2)	55 (32.9)	309 (29.4)	0.01 ^c^	0.31 ^e^
Heart rate mean (SD)	91.9 (23.9)	88.1 (23.3)	91.0 (23.8)	0.03 ^a^	
Systolic blood pressure mean (SD)	123.0 (29.4)	141.2 (37.8)	127 (32.5)	0.01 ^d^	
Shock index	0.8 (0.6)	0.6 (0.4)	0.7 (0.6)	0.01 ^a^	
Respiratory rate mean (SD)	20.0 (10.5)	18.3 (7.5)	19.6 (9.9)	0.03 ^a^	
Oxygen saturation mean (SD)	93.7 (15.2)	91.2 (17.9)	93.1 (15.8)	0.01 ^a^	
GCS > 8 *n* (%)	556 (69.0)	135 (55.1)	691 (65.8)		0.01 ^e^
GCS ≤ 8 *n* (%)	250 (31.0)	110 (44.9)	360 (34.2)	0.01 ^b^	0.01 ^e^
Injury Severity Score <16	154 (19.1)	47 (20)	201 (19.1)		0.89 ^e^
Injury Severity Score ≥16	652 (80.9)	198 (80)	850 (80.9)		0.78 ^e^
NACA score ≥4 *n* (%)	783 (97.1)	243 (99.2)	1026 (97.6)		0.06 ^e^
Trauma type *n* (%)					
Road crash	392 (48.6)	55 (22.5)	447 (42.5)	0.01 ^c^	0.01 ^e^
Falls	278 (34.5)	150 (61.2)	428 (40.7)		0.01 ^e^
Other	86 (10.7)	15 (6.1)	101 (9.6)		0.58 ^e^
Pedestrian hit	48 (6.0)	22 (9.0)	70 (6.7)		0.64 ^e^
Assault	1 (0.1)	3 (1.2)	4 (0.4)		NA
Sports injury	1 (0.1)	0	1 (0.1)		NA
Comorbidities *n* (%)					
0	556 (70.2)	56 (22.4)	622 (59.1)		0.01 ^e^
1	169 (21.0)	70 (28.6)	239 (22.7)		0.21 ^e^
2+	71 (8.8)	120 (49.0)	191 (18.2)		0.01 ^e^
Anticoagulant therapy *n* (%)	7 (0.9)	45 (18.4)	52 (4.9)	0.01 ^c^	
Anti-platelet therapy *n* (%)	18 (2.2)	51 (20.8)	69 (6.6)	0.01 ^c^	
ICU stay (days) mean (SD)	4.5 (7.2)	3.4 (5.9)	4.2 (6.9)	0.24 ^a^	
Hospital stay (days) mean (SD)	17.4 (22.0)	10.9 (15.7)	15.9 (20.8)	0.01 ^a^	
Hospital mortality	112 (13.9)	114 (46.5)	226 (21.5)	0.01 ^a^	
48 h mortality	80 (9.9)	82 (33.6)	162 (15.4)	0.01 ^a^	

Note: ED = Emergency department; NACA = National Advisory Committee for Aeronautics; SD = Standard deviation; ^a^ Wilcoxon–Mann–Whitney test; ^b^ Chi-squared test; ^c^ Fisher’s exact test; ^d^ Student’s t-test; ^e^ Z test; NA: not applicable.

**Table 2 jcm-09-01343-t002:** Comorbidities.

	<65 Years	≥65 Years	Total	*p*-Value
Psychiatric + Addiction	138 (39.2)	44 (10.7)	182	0.01 ^a^
Cardiologic	69 (19.6)	142 (34.6)	211	0.01 ^a^
Oncologic	13 (3.6)	43 (10.4)	56	0.01 ^a^
Pulmonary	23 (6.5)	34 (8.2)	57	0.07
Musculoskeletal	35 (9.9)	28 (6.8)	63	0.38 ^a^
Neurologic	39 (11)	53 (12.9)	92	0.15 ^a^
Metabolic	35 (9.9)	66 (16)	101	0.01 ^a^

Note: ^a^ Z test.

**Table 3 jcm-09-01343-t003:** Hospital mortality.

Logistic regression				Number of observations	643		
				LR chi2 (14)	300.82		
				Prob > chi2	0.01		
Log likelihood = −176.00				Pseudo R2	0.46		
Hospital mortality		Odds Ratio	Standard Error	z	*p* > z	(95% Confidence Interval)	
age		1.06	0.01	6.75	0.01	1.04	1.08
NACA		2.23	0.66	2.70	0.01	1.25	3.99
GCS		0.83	0.029	−5.12	0.01	0.78	0.89
SpO2		1.00	0.006	3.26	0.01	1.00	1.00
ISS		1.08	0.01	5.50	0.01	1.05	1.10
anti-platelet		3.22	1.67	2.25	0.02	1.16	8.91

**Table 4 jcm-09-01343-t004:** Injury Patterns (*n* = 1051).

n (%)	≥65 Years	<65 Years	Total
Brain *	150 (14.3)	342 (32.5)	492 (46.8)
Thoracic	76 (7.2)	332 (31.6)	408 (38.8)
Lower limbs	50 (4.8)	296 (28.1)	346 (32.9)
Facial/ENT	50 (4.8)	201 (19.1)	251 (23.9)
Other	49 (4.7)	171 (16.2)	220 (20.9)
Spine	42 (4.0)	235 (22.3)	277 (26.3)
Rib fracture	25 (2.4)	64 (6.1)	89 (8.5)
Upper limbs	25 (2.4)	198 (18.8)	223 (21.2)
Abdominal	15 (1.4)	149 (14.2)	164 (15.6)
Cervical fracture	15 (1.4)	40 (3.8)	55 (5.2)
Pelvic fracture	13 (1.2)	53 (5.1)	66 (6.3)
Lumbar fracture	12 (1.1)	67 (6.4)	79 (7.5)
Femoral fracture	10 (0.9)	57 (5.4)	67 (6.3)
Clavicular fracture	7 (0.6)	27 (2.6)	34 (3.2)
Cardiac arrest	7 (0.6)	30 (2.9)	37 (3.5)
Pneumothorax	6 (0.6)	40 (3.8)	46 (4.4)
Facial fracture (orbit, petrosal and frontal bone)	6 (0.6)	37 (3.5)	43 (4.1)
Tibial fracture	4 (0.4)	16 (1.5)	20 (1.9)
Liver rupture	4 (0.4)	22 (2.1)	26 (2.5)
Humeral fracture	4 (0.4)	20 (1.9)	24 (2.3)
Pulmonary contusion	4 (0.4)	27 (2.6)	31 (2.9)
Splenic rupture	3 (0.3)	45 (4.3)	48 (4.6)
Wrist fracture	3 (0.3)	16 (1.5)	19 (1.8)
Dorsal thoracic fracture	3 (0.3)	21 (2.0)	24 (2.3)
Maxillary fracture	2 (0.2)	14 (1.3)	16 (1.5)
Mandibular fracture	1 (0.1)	9 (0.8)	10 (0.9)

* Brain Injury includes subdural, subarachnoid, epidural and parenchymal hemorrhage. Other includes cardiac arrest, contusion, wound. ENT: Ear, nose and throat.

**Table 5 jcm-09-01343-t005:** Immediate treatment (*n* = 1051).

*n* (%)	<65 Years	≥65 Years	Total	*p*-Value
Intervention	591 (73)	106 (43)	697 (66.3)	
Orthopedic intervention	243 (30.1)	26 (10.6)	269 (25.5)	0.01 ^a^
Neurosurgical intervention	186 (23)	49 (20)	235 (22.3)	0.31 ^a^
Abdominal intervention	54 (6.7)	8 (3.2)	62 (5.8)	0.04 ^a^
Cardiopulmonary resuscitation	27 (3.3)	9 (3.6)	36 (3.4)	0.81 ^a^
Cardiac intervention	18 (2.2)	2 (0.8)	20 (1.9)	0.12 ^b^
Embolization	57 (7)	10 (4)	67 (6.3)	0.10 ^b^
ENT intervention	6 (0.7)	2 (0.8)	8 (0.7)	0.98 ^b^
None	290 (37)	140 (57)	430 (40.9)	0.01 ^a^

Note: ^a^ Chi-squared test; ^b^ Fisher’s exact test.

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
