# Peer review of "Retrospective Analysis of Geriatric Major Trauma Patients Admitted in the Shock Room of a Swiss Academic Hospital: Characteristics and Prognosis"

_jcm, 2020, doi:10.3390/jcm9051343_

Round 1

Reviewer 1 Report

Thank you for the opportunity to review this interesting and well-conducted retrospective observational study. The study provides several new insights into an under-researched area of practice and the methods and results are overall, very clearly outlined and presented. There are a small number of minor edits and clarifications that would further strengthen this paper, and increase its relevance and reach to a non-clinical audience, which I have outlined below. Thank you for undertaking this work and furthering this important field of care.

  • Introduction/Aim: It would be helpful to clarify in your given aim that you are comparing the older to younger population. The abstract and intro suggest both – Lines 14-15, 46-49, 62-63. The inclusion criteria begins at age 16 so putting this into the given aim would improve clarity.
  • You acknowledge that the literature on this topic is sparse but it is important to highlight that it is also quite old (which further highlights why this present study is needed).
  • Methods: To a non-specialist it is not clear why you selected the given exclusion criteria of penetrating injuries & burns.
  • The studies uses the NACA score – to a non-specialist it would be useful to have some discussion as to why this tool was used (as it is not immediately clear, as with the ISS and GCS). Possibly a short sentence could be added in lines 93-94 to explain why it was selected.
  • Results: Table 4 refers to pattern of injury but presentation of information (the order the information is given in) does not make it immediately clear what the patterns are. Unless there is some logic to the author presentation that is not immediately obvious it would be helpful to see the list in order of frequency (probably for older people as they are your key population).
  • Discussion: you reference several other studies finding similar outcomes to you. Given that you highlight the uniqueness of the Swiss location, and the prevalence of data from the US, it would be interesting and useful to give greater discussion on variances or similarities based on location. This would help see where there are wider generalizations that can be made internationally.
  • Lines 213-214, the reasons seem probable but it would be helpful to have some supporting literature to back this from Switzerland (given how frequently advance directives are not found/not followed/not used in other countries, e.g. the US and UK).
  • Some differences in your findings may also be contributed to the fact that existing comparative data are in many cases, much older and so changes in care provision, drug refinement, and service restructures may also be a factor to consider. In addition – are those data from other countries – this point was raised in the introduction and again, seems like an important one to address.
  • This article might prove helpful as it gives some recent European data comparing older to younger trauma: Verbeek, D.O., Ponsen, K.J., Fiocco, M. et al. Pelvic fractures in the Netherlands: epidemiology, characteristics and risk factors for in-hospital mortality in the older and younger population. Eur J Orthop Surg Traumatol 28, 197–205 (2018). https://doi-org.proxy1.library.jhu.edu/10.1007/s00590-017-2044-3
  • Limitations: limitations are clearly stated, but it would be helpful to see some of the rationale for why all tests were not performed.

Reviewer 2 Report

This is a very interesting study on outcome of severely injured patients of over 65 years of age. As These important data add to the already available knowledge, they merit to be published.

The only criticism from my side is the insufficient discussion of the factor "underestimation of the severity of injury in old persons". Have the authors adapted their diagnostic and treatment protocols for persons of old age, or do that think that these outcomes cannot by influenced by any protocol?

Specifically:

Lines 209-211: 57% of the G>65 group did not receive an immediate treatment (Table 5). Did the delay in treatment influence morbidity and mortality in G>65 group? Can underestimation of the severity of injury play a role for delay of immediate treatment? This is also visible in the % of medicalised ambulance (Table 1), which is significantly lower in G>65 than in G<65. Please also consider the following references in discussing this.

Phillips S, Rond 3rd PC, Kelly SM, Swartz PD. The failure of triage to identify geriatric patients with tharauma: results of the Florida Tauma Triage Study. J Trauma 1996;40:278

Zimmer-Gembeck MJ, Southard PA, Hadges JR, Mullins RJ, Rowland D, Stone JV, Trunkey DD. Triage in an estabished trauma System. J Trauma 1995;39(5):922-8

Sheetz LJ. Trends in the accuracy of older persons trauma triage from 2004 to 2008. Prehosp Emerg Care 2011;15:83-7

Rommens PM, Kuhn S. Principles of damage control in the elderly. In "Damage control in the polytraumatized patient" second Edition . Pape HC, Peitzman AB, Rotondo MF, Giannoudis PV. Eds. Springer International Publishing AG 2017: pp. 249-261. 
